# Evaluating the effects of community treatment orders (CTOs) in England using the Mental Health Services Dataset (MHSDS): protocol for a national, population-based study

Scott Weich,[1] Craig Duncan,[2] Kamaldeep Bhui,[3] Alastair Canaway,[4] David Crepaz-Keay,[5] Patrick Keown,[6] Jason Madan,[4] Orla McBride,[7] Graham Moon,[8] Helen Parsons,[4] Swaran Singh,[4] Liz Twigg[2]

For numbered affiliations see end of article.

**Correspondence to**
Dr Craig Duncan;
craig.duncan@port.ac.uk

## ABSTRACT

**Introduction** Supervised community treatment (SCT) for people with serious mental disorders has become accepted practice in many countries around the world. In England, SCT was adopted in 2008 in the form of community treatment orders (CTOs). CTOs have been used more than expected, with significant variations between people and places. There is conflicting evidence about the effectiveness of SCT; studies based on randomised controlled trials (RCTs) have suggested few positive impacts, while those employing observational designs have been more favourable. Robust population-based studies are needed, because of the ethical challenges of undertaking further RCTs and because variation across previous studies may reflect the effects of sociospatial context on SCT outcomes. We aim to examine spatial and temporal variation in the use, effectiveness and cost of CTOs in England through the analysis of routine administrative data.

**Methods and analysis** Four years of data from the Mental Health Services Dataset (MHSDS) will be analysed using multilevel models. Models based on all patients eligible for CTOs will be used to explore variation in their use. A subset of CTO-eligible patients comprising a treatment group (CTO patients) and a matched control group (non-CTO patients) will be used to examine variation in the association between CTO use and study outcomes. Primary outcome will be total time in hospital. Secondary outcomes will include time to first readmission and mortality. Outputs from these models will be used to populate predictive models of healthcare resource use.

**Ethics and dissemination** Ethical approval has been granted by the National Health Service Data Access and Advisory Group and Warwick University. To ensure patient confidentiality and to meet data governance requirements, analyses will be carried out in a secure microdata laboratory using de-identified data. Study findings will be disseminated through academic channels and shared with mental health policy-makers and other stakeholders.

### Strengths and limitations of this study

► This will be the largest and most complete study of its kind and the national representativeness of the study sample, deriving from routine clinical activity, is a major strength.
► The study design reflects the range of real-world settings in which mental health services are delivered and patients live in England.
► The use of multilevel models will allow us to estimate variation in community treatment order (CTO) use and outcome at individual, area and service level and to establish who CTOs work for and in what circumstances.
► Study limitations include confounding by indication and bias arising from missing data within the routine administrative data source.

## INTRODUCTION

Community treatment orders (CTOs), a form of supervised community treatment (SCT), were introduced to the Mental Health Act (2007) (MHA) in England and Wales in 2008 and allow certain patients detained in hospital to continue treatment in the community following discharge under specific conditions. To be eligible for a CTO, a patient must be detained in hospital under the MHA on a section that allows for compulsory treatment. The most common inpatient treatment orders are sections 3 and 37, the former being a civil order and the latter a forensic, court-imposed equivalent for mentally ill offenders. A CTO provides the clinical team the power to recall the patient to hospital after discharge.

At CTO initiation (the point of discharge from inpatient care), there are two mandatory conditions, namely that the person subject to a CTO makes themselves available

for assessment for its renewal and (where necessary) to be seen by an approved second opinion doctor (SOAD) for the purposes of assessing proposed treatment plan should the patient lack capacity in respect of their treatment. Discretionary conditions can also be specified which most often include treatment (ie, medication) adherence and engagement with services (eg, attending for appointments with professionals). CTOs do not permit compulsory (ie, forcible) treatment in the community,[1] but those who default from these conditions may be recalled to hospital for assessment if there are grounds to suspect deterioration in their mental health and/or risk to the patient or others. Where this assessment indicates the need for further treatment, the CTO is revoked and the original treatment order applies.

There were two motivations for introducing CTOs in England and Wales[2]: to reduce 'revolving door' admissions associated with non-adherence with care by a small group of patients, and to allow treatment in the least restrictive setting, in accordance with the Human Rights Act (1988). While it was originally envisaged that around 10% of eligible patients would be placed on CTOs, the figure is currently closer to 30%.[3] Since more CTOs are issued than are ended each year, the number of patients subject to CTOs has increased over time. Almost five thousand CTOs were issued in 2016–2017, with use varying between demographic groups: rates of CTO use for males (11.4 per 100 000 population) were almost twice the rate for females (6.6 per 100 000 population), and the rate for black and black British people (60.1 per 100 000 population) was almost nine times the rate for those of white ethnicity (6.8 per 100 000 population).[4] Use also varies between National Health Service (NHS) (provider) Trusts.[5 6]

SCT has been adopted in over 75 other jurisdictions around the world,[7] although it is used differently in different places. This is highlighted by its different names, including 'involuntary outpatient commitment', 'assisted outpatient treatment', 'supervised discharge', 'compulsory community treatment' and 'compulsory ambulatory treatment'.[8] In some places, SCT is initiated only by courts, while elsewhere, as in England, it remains a matter exclusively for health and social care providers, although within a framework prescribed by statute and subject to legal review. There are also differences between countries in the organisation and funding of mental health services, including separation of inpatient from community mental health teams and variations in the nature and availability of care for people with severe mental illness. Generalising findings between countries is therefore problematic.

Despite widespread use, SCT remains controversial. In part, this is because patients and carers see it as coercive, raising fears of negative effects on therapeutic relationships.[9] This disquiet has been heightened by conflicting research evidence about effectiveness.[10] There have been only three randomised controlled trials of CTOs, two in the USA and one in the UK. None demonstrated significant benefit of CTOs in terms of hospital readmission, social functioning, quality of life, mental state or offending.[11–13] Randomised trials of CTOs face particular ethical and practical challenges (including relatively brief duration of follow-up), and these limit their utility.[14] In critically appraising the two US trials, Kisely *et al* highlighted small sample sizes, the court-imposed nature of compulsory community treatment and the exclusion of patients with a history of violence. The OCTET trial in England experienced several methodological limitations, including the involuntary nature of the control condition, high rates of non-participation and patients who moved between trial arms after randomisation.[15–17]

Apart from the inevitable unblinding of participants and clinicians, it is impossible to disentangle the effects of legal compulsion from the vigilance and intensive care associated with CTO implementation.[8] Further trials of CTOs are unlikely, given ethical challenges inherent in randomising those who may lack insight, and who may be dangerous to themselves or others, to legal restriction versus truly voluntary care.[18]

The results of observational studies are less consistent, with some showing decreased admission rates and reduced bed use among those subject to SCT, while others do not.[2 8 10 14 19] After excluding uncontrolled studies, observational research is characterised by matched cohort designs and before and after studies, with outcomes generally (but not invariably) assessed over 2 years or less. More recently, findings have emerged which suggest that patients on CTOs may experience better physical health and reduced mortality.[20 21]

Population-based observational studies using routine clinical and administrative data allow analysis of variation in SCT use and outcome to be studied.[10 22] We will use 4 years of data from the Mental Health Services Dataset (MHSDS) to address the following aims: (1) to examine spatial and temporal variation in the use of CTOs; (2) to estimate associations between CTO use and patient outcomes; (3) to explore variation in these associations between patients, places and providers, establishing when, where and for whom CTOs may be effective and (4) to estimate the financial costs and benefits associated with CTOs.

## METHODS AND ANALYSIS
### Data sources and study outcomes

The use of CTOs in England is recorded as part of a mandatory administrative dataset, the MHSDS, formerly known as the Mental Health Minimum Dataset, and then the Mental Health and Learning Disabilities Dataset.[23] MHSDS collates monthly returns from health service providers on all patients in contact with secondary mental health services provided and/or funded by NHS England. This includes voluntary and involuntary inpatient treatment, outpatient attendance, day treatment and other episodes of secondary mental healthcare.

**Table 1** Spatial/service setting identifiers present in the Mental Health Services Dataset

| Spatial/service setting identifier | Description |
|---|---|
| Lower layer super output area | Local area of residence, based on 2001 Census boundaries, typically containing 672 households and 1614 residents |
| General practitioner practice | Primary care provider |
| National Health Service and independent sector provider trust | Provider of secondary mental healthcare |
| Primary care trust | Commissioner of secondary mental healthcare |

MHSDS data are based on spells of care for individual patients. Since 2011, the dates on which specific care spells start and end have been included. Care spell dates can therefore be used to determine duration of CTO use and hospital (re)admissions. Four years of MHSDS data will be used covering the period 2011/2012 to 2014/2015, hereafter referred to as the study period. MHSDS records contain a unique patient identifier code which will be used to link patient care spells across the study period. This identifier will also be used to link patient records to the Office for National Statistics (ONS) Mortality Database, to ascertain patient deaths. Access to and linkage of, these different administrative data sets will be facilitated by the ESRC Administrative Data Research Network.

### Covariates

MHSDS provides data on a range of patient characteristics, including age, sex and ethnicity. Data on clinical characteristics are also available such as diagnosis, care clusters (groupings of patients with similar needs and problem severities)[24] and Health of the Nation Outcome Scales (outcome ratings for a range of health and social domains).[25] Additional patient characteristics such as marital, employment and accommodation status may be

present in the dataset, although previous research noted high levels of missing data.[26]

MHSDS patient records also include several spatial/ service setting identifiers (table 1). These will be used to link patient records to data on factors shown by previous studies to influence mental health outcomes at local area and service provider level. Specifically, English Indices of Deprivation 2015 will provide information about local area socioeconomic characteristics while ONS 2011 Census data will be used to characterise local populations, including through the use of population density and ethnic composition data.[27 28]

### Data analysis

As the overarching aim of the study is to examine spatial and temporal variation in CTO use and outcome, multi-level models (MMs) will be used.[29] MMs allow variation in outcomes to be apportioned to the appropriate spatial level (eg, between patients, local areas and service settings) as well as estimating associations with individual, area and service-level characteristics.[30] Since the local areas that patients live in do not nest hierarchically within the health service settings where they receive care (and vice versa), cross-classified MMs (CCMMs) will be used (figure 1).[31]

Analysis will be conducted in three stages. First, CCMMs based on all patients eligible for CTOs will be used to explore spatial and temporal variation in the use of CTOs. CTO use will be considered as a binary outcome (patients subject to a CTO during the study period vs those who were not) and analyses will be undertaken using logistic multilevel regression. Models will estimate the variation at each level in the risk of being on a CTO and assess the extent to which any variation in this outcome is associated with patient, local area and health service characteristics. We will also investigate the duration of time spent subject to CTO (in days). Poisson/negative binomial link function CCMMs will be used and will include offset variables to take account of different observation times for patients.[32]

Next, to examine associations between CTO use and a range of outcomes, a cohort of patients will be

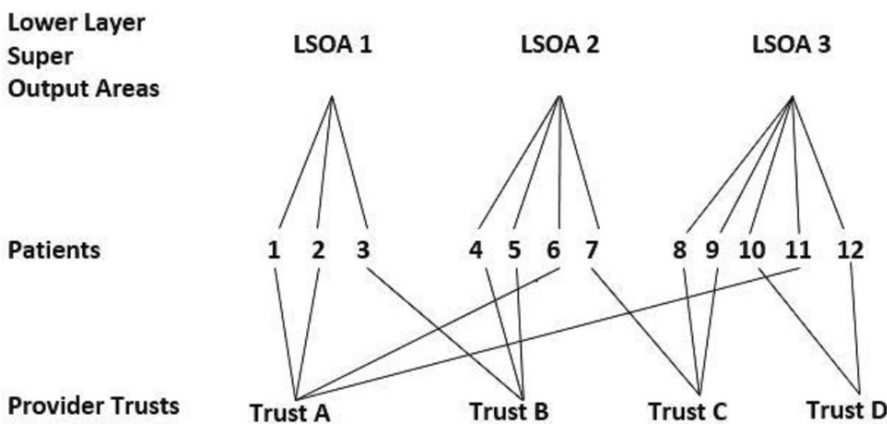

**Figure 1** An illustration of cross-classified multilevel data.

constructed consisting of a treatment group (those who have been on a CTO) and a matched control group (those who are eligible but who have not been on a CTO). To prevent bias arising from secular changes in clinical practice, we will frequency match these groups according to the start date of the CTO (CTO group) and the end date of the CTO-eligible section of the MHA (non-CTO group), with tolerance limits determined after inspection of the data. As these dates are the point within the study period at which patients in both treatment groups are discharged from hospital, they represent the index date for each arm of the study. Models based on the cohort of matched patients will be used to test associations between SCT use and patient outcomes: total time spent in hospital, time to first readmission and mortality. For total time spent in hospital, count models based on Poisson or negative binomial distributions will be estimated. For time to first readmission, discrete-time survival analysis models will be estimated. For mortality, binary response models based on the logistic distribution will be estimated. To control for confounding by indication, we will adjust analyses for the propensity of being placed on SCT,[33] with propensity scores being modelled using the pool of available variables which will include age, sex, ethnicity, diagnosis and previous admissions (where known). CCMMs based on this cohort will be used to test for the overall 'average' effect of CTO on each of the study outcomes. The models will also assess whether there is significant spatial and temporal variation around these average effects and estimate the extent to which this is associated with patient, local area and health service characteristics, including the length of time that patients are on CTOs.

Finally, we will use outputs from our CCMMs to populate predictive models of healthcare resource use incorporating the predicted impact of CTOs. We will model differences in treatments costs (eg, bed days and number of contacts with mental health staff) between patients who are and are not on CTOs. We will also quantify the administrative burden of CTOs, using protocols for mandated governance processes (eg, Mental Health Act Review Tribunals and SOADs) that apply to the use of CTOs. Projected costs associated with different plausible future trajectories of CTO use will be calculated and a probabilistic sensitivity analysis will be presented for each projection based on estimates of parameter uncertainty derived from the CCMMs.

CCMMs will be estimated in MLwIN using Markov chain Monte-Carlo (MCMC) Bayesian methods.[34] This approach is suited to the complexity of the MHSDS and will provide unbiased measures of variation in CTO use and outcome, and associations with patient, local area and health setting characteristics. MCMC modelling diagnostics, together with measures of multilevel variation, will be used to evaluate different models to quantify where, and for whom, associations and variation occurs.[35–37]

## PATIENT AND PUBLIC INVOLVEMENT

Two reference groups will provide guidance to the project to maximise its relevance and impact. The first group will consist of mental health service users and carers and will include people with lived experience of being subject to CTOs or of caring for someone who has. Participants will be recruited by the Mental Health Foundation, the largest mental health policy, research and service improvement charity in England. Recruitment will ensure diversity of age, gender and ethnicity. The second group will consist of mental health professionals and will include managers and clinicians. The reference groups will help us ensure that analyses and dissemination are relevant to the needs of stakeholders.

## ETHICS AND DISSEMINATION

To ensure patient confidentiality and to meet data governance requirements, analyses will be carried out in a secure microdata laboratory at the ONS using de-identified data. Reporting of the study will be consistent with the Strengthening the Reporting of Observational Studies in Epidemiology guidelines.[38]

CTOs are used more often than originally anticipated in England. Although many clinicians continue to see their value,[39] uncertainty and concern about their use has been expressed by regulators and legislators.[3 40] Both have called for the sort of nuanced evidence about the benefits and hazards of CTO use in real-world settings which can only come from large-scale, population-based, observational research. The purposes of this study are to describe the practice in respect of CTO use and duration, to estimate associations with, and spatial variation in, outcomes, and to understand where, and for whom, CTOs might be of benefit. We will also model the economic costs and benefits of current patterns of CTO use.

This will be the largest and most complete study of its kind in England. The national scope and representativeness of the study sample, deriving from routine clinical activity, is a major strength, as is the duration of follow-up. Study limitations are those that affect observational and secondary research, particularly confounding by indication, residual confounding and bias. Issues of data quality (including missing data) are important sources of the latter.[41] The study findings will be of direct policy relevance, and we plan to share them directly with those bodies working in this area at the earliest opportunity.

### Author affiliations
[1]School of Health and Related Research, University of Sheffield, Sheffield, UK
[2]Department of Geography, University of Portsmouth, Portsmouth, UK
[3]Centre for Psychiatry, Barts and The London School of Medicine & Dentist, University of London, London, UK
[4]Warwick Medical School, University of Warwick, Coventry, UK
[5]Mental Health Foundation, London, UK
[6]Institute of Neuroscience, Newcastle University, Newcastle upon Tyne, UK
[7]School of Psychology, Ulster University, Londonderry, UK
[8]Geography and Environment, Ulster University, Southampton, UK

**Contributors** SW, KB, DC-K, PK, JM, OM, GM, HP, SS and LT had the original idea for the study and were responsible for study hypotheses, design and data

specification. AC contributed to the design of the health economics work. CD was responsible for drafting the manuscript. SW had overall responsibility for the conduct of the study. All authors contributed, read and approved the final manuscript.

**Funding** The work was supported by the National Institute for Health Research Health Services and Delivery Research Programme (project number 14/52/40).

**Competing interests** None declared.

**Patient consent** Not required.

**Ethics approval** Ethical approval has been granted by the NHS Data Access and Advisory Group and the University of Warwick's Biomedical and Scientific Research Ethics Committee (BSREC Ref: REGO-2015–1623).

**Provenance and peer review** Not commissioned; externally peer reviewed.

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
