## [Reviewer comments · BMJ Open]

ARTICLE DETAILS

TITLE (PROVISIONAL)	Evaluating the effects of Community Treatment Orders (CTOs) in England using the Mental Health Services Dataset (MHSDS): Protocol for a national, population-based study
AUTHORS	Weich, Scott; Duncan, Craig; Bhui, Kamaldeep; Canaway, Alastair; Crepaz_Keay, David; Keown, Patrick; Madan, Jason; McBride, Orla; Moon, Graham; Parsons, Helen; Singh, Swaran; Twigg, Liz

VERSION 1 – REVIEW

REVIEWER	Steve Kisely University of Queensland, Australia
REVIEW RETURNED	23-May-2018

GENERAL COMMENTS	This study will use 4 years of England-wide administrative data (the MHSDS) to explore the use of CTOs. Outcomes will be compared with controls. Clinical characteristics that are available include diagnosis, care clusters (groupings of patients with similar needs and problem severities) and scores from the Health of the Nation Outcome Scales. The MHSDS potentially also collects information on marital, employment and accommodation status although previous research has noted high levels of missing data. This will be a major limitation when constructing the control group MHSDS patient records also include several spatial/service setting identifiers. These will be used to link patient records to data on factors shown by previous studies to influence mental health outcomes at local area and service provider level. Models will estimate the variation at each level in the risk of being on a CTO and assess the extent to which any variation in this outcome is associated with patient, local area and health service characteristics. In their comparison with controls, they will adjust analyses for the propensity of being placed on CTOs. The models will also assess whether there is significant spatial and temporal variation around these average effects and estimate the extent to which this is associated with patient, local area and health service characteristics. Unfortunately, they do not describe in detail what sort of variables they will include in their propensity score. Will this include pre-CTO health service use, which is an important predictor of subsequent use? In addition, what is the index date? Is it discharge from hospital for cases and controls or the start of a new care episode? Does the index date for each case & control have to be within a certain period? I also couldn't see anything about the effect of length of CTO placement. Finally, although they note that previous studies have looked at the effect of CTOs on mortality, they do not propose to look at this outcome in the [resent study. Is it possible to link these data to mortality data?
--

	Strengths of the study are nation-wide coverage, thereby eliminating section bias and reflecting real- world use. Limitations are lack of detail of what will be included in the propensity score, as well as missing information on important confounders. On a small note, the introduction on page 8 , lines 39 -51, could reflect more accurately that while some observational studies have shown decreased admission rates and reduced bed use among CTO cases, others have not
--	---

REVIEWER	Ruth Vine Melbourne Health, Australia
REVIEW RETURNED	26-May-2018

GENERAL COMMENTS	This study proposes using a large data base to interrogate various facets of the use of CTO, including differences found in frequency of use across health services. Given the challenges in doing RCT in this area, and the strongly held competing ideologies regarding CTO, a study of this size that permits a multilevel model analysis of data that is routinely collected across all services provides a strong methodology highly likely to yield solid results of value to clinicians and policy makers. The proposed data to be used covers 2011/12 to 2014/15. Ideally the study would use more recent data given shifts in population and bed availability, but I appreciate that getting such large data sets can be difficult to obtain. The statistical analysis proposed appears appropriate but a specialist statistical review may be indicated.
---

VERSION 1 – AUTHOR RESPONSE

Response to reviewers' comments

We are grateful for the opportunity to address the comments made by the reviewers. Our responses and accompanying changes to the text are as follows:

Reviewer 1:

1. Unfortunately, they do not describe in detail which variables they will include in their propensity score. Will this include pre-CTO health service use, which is an important predictor of subsequent use?

RESPONSE: We provide detail of the variables that will be included in the propensity scores through the inclusion of the following text on page 12:

AMENDED TEXT "..., with propensity scores being modelled using the pool of available variables which will include age, sex, ethnicity, diagnosis and previous admissions (where known)."

2. In addition, what is the index date? Is it discharge from hospital for cases and controls or the start of a new care episode? Does the index date for each case and control have to be within a certain period?

RESPONSE: The time of entry into the study cohort is taken as the start date of the CTO (which coincides with discharge from hospital for the exposed group), and discharge from hospital (for those eligible but not placed not on CTOs).

We have modified some of the existing text on page 11 and added a new sentence to clarify the index date for patients in both arms of the study:

AMENDED TEXT: "To prevent bias arising from secular changes in clinical practice, we will frequency-match these groups according to the start date of the CTO (CTO group) and the end date of the CTO-eligible section of the Mental Health Act (non-CTO group), with tolerance limits determined after inspection of the data. As these dates are the point within the study period at which patients in both treatment groups are discharged from hospital, they represent the index date for each arm of the study."

3. I couldn't see anything about the effect of length of CTO placement.

RESPONSE: We are grateful to the reviewer for drawing our attention to this oversight. We omitted this from our original manuscript in error but have included it by extending a sentence on page 12:

AMENDED TEXT: "The models will also assess whether there is significant spatial and temporal variation around these average effects and estimate the extent to which this is associated with patient, local area and health service characteristics, including the length of time that patients are on CTOs."

4. Finally, although they note that previous studies have looked at the effect of CTOs on mortality, they do not propose to look at this outcome in the present study.

RESPONSE: In fact, we do intend to look at this outcome and on page 9 of the original manuscript we outlined how mortality data will be linked to MHSDS data while on page 12 we outlined the analyses of mortality that will be undertaken. These sections remain in the revised version.

5. On a small note, the introduction on page 8, lines 39-51, could reflect more accurately that while some observational studies have shown decreased admission rates and reduced bed use among CTO cases, others have not.

RESPONSE: This is a very valid point. We have therefore amended a sentence and added a reference to reflect this point on page 8:

AMENDED TEXT: "The results of observational studies are less consistent, with some showing decreased admission rates and reduced bed use among those subject to SCT, while others do not.^{2,8,10, 14,19}"

Reviewer 2:

1. Ideally the study would use more recent data given shifts in population and bed availability.

RESPONSE: We agree with this point but unfortunately 2014/15 was the latest year for which data were available when negotiations for data access took place, given the funding timeline of the project.

VERSION 2 – REVIEW

REVIEWER	Steve Kisely University of Queensland
REVIEW RETURNED	19-Jun-2018
GENERAL COMMENTS	The authors have responded to all r/wers' concerns. One additional suggestion is reference to the STROBE guidelines.

VERSION 2 – AUTHOR RESPONSE

Response to reviewers' comments

We are grateful for the opportunity to address the latest comments made by the reviewer. Our response and accompanying changes to the text are as follows:

Reviewer 1:

1. *One additional suggestion is reference to the STROBE guidelines.*

RESPONSE: We refer to the STROBE guidelines through the inclusion of the following text on **page 13** together with a new reference on **pages 19-20** :

AMENDED TEXT: "Reporting of the study will be consistent with the STROBE guidelines.³⁸"

"38 von Elm E, Altman DG, Egger M, Pocock SJ, Gøtzsche PC, Vandembroucke JP, STROBE Initiative. The Strengthening the Reporting of Observational Studies in Epidemiology (STROBE) statement: guidelines for reporting observational studies. *Ann Intern Med* 2007 Oct 16;147(8):573-577."